# Nutrition Quality of Packaged Foods in Bogotá, Colombia: A Comparison of Two Nutrient Profile Models

**DOI:** 10.3390/nu11051011

**Published:** 2019-05-04

**Authors:** Mercedes Mora-Plazas, Luis F. Gómez, Donna R. Miles, Diana C Parra, L. S. Taillie

**Affiliations:** 1Departamento de Nutrición Humana, Universidad Nacional de Colombia, Bogotá, Carrera 45 N°26-85, Bogotá 11001, Colombia; 2Facultad de Medicina, Pontificia Universidad Javeriana, Bogotá, 8 piso Hospital Universitario San Ignacio, Bogotá 110231, Colombia; L.gomezg@javeriana.edu.co; 3Carolina Population Center, University of North Carolina, Chapel Hill, NC 27599, USA; drmiles@email.unc.edu; 4Program of Physical Therapy, Washington University in St. Louis School of Medicine, 4444 Forest Park Avenue, St. Louis, MO 63108, USA; parrad@wustl.edu; 5Carolina Population Center and Department of Nutrition, Gillings School of Global Public Health, University of North Carolina, Chapel Hill, NC 27599, USA; taillie@unc.edu

**Keywords:** front-of-package labels, warning labels, labelling, nutrient profile models, Latin America, food policy, sugar-sweetened beverages, junk food

## Abstract

Background: Front-of-package warning labels are an increasingly common policy and have been implemented to inform consumers of the nutritional quality of ultra-processed foods. This study examined the proportion of Colombian products that could be subjected to such regulations. Methods: Two nutrition profile models were compared: the Pan American Health Organization (PAHO) model, and the nutrient profile established under the Chilean food labeling and advertising law (Chilean model). Products (*n* = 6708) exceeding nutrient criteria based on each model were identified as subject to regulation. Results: A total of 80.2% (PAHO model) to 66.4% (Chilean model) of foods met the criteria for regulation. The categories with the highest proportion of regulated products were meats (97.3% PAHO model; 87.5% Chilean model), sweets (95.6% PAHO) and snacks (Chilean model). The category with the lowest proportions of regulated products were cereals (47.3% PAHO model) and miscellaneous foods and fish/seafood (39.0% and 39.5%, respectively, Chilean model). Conclusions: Under both the PAHO and Chilean nutrient profile models, the majority of packaged foods available in Bogotá would be eligible to receive front-of-package warning labels. These results suggest a warning label law could have a major impact on the Colombian food supply, especially in the context of the growing prevalence of diet-related chronic diseases in Colombia.

## 1. Introduction

In 2017, the second leading risk factor associated with mortality and disability-adjusted life-years in Colombia was attributed to unhealthy dietary patterns [1]. Such diets often consist of ultra-processed foods and beverages with high caloric densities in conjunction with high levels of added sugars, saturated fats, and sodium, while providing low levels of vitamins and minerals [2]. Compelling evidence indicates that the consumption of ultra-processed foods and beverages is associated with increased risk of obesity, type 2 diabetes, hypertension, and weight gain, among other conditions [3,4,5,6,7]. From 2000 to 2013, a significant increase in consumer sales of ultra-processed foods and beverages has been observed among all Latin American countries, with the exception of Venezuela and Argentina [8]. In Colombia specifically, the consumption of ultra-processed products, such as pre-packaged desserts, confectionary, salty snacks, snack bars, cakes and pastries, has risen dramatically from 2005 to 2017 [9].

In response to both the continued rise in obesity and increased consumption of ultra-processed products, policies aimed at informing consumers of foods and beverages which are high in calories, sugar, saturated fat and sodium are being developed with the intention of reducing the intake of these products. A number of countries in Latin America have proposed or implemented front-of-package (FOP) warning labels, which require packaged foods with nutrient values over set thresholds to include specified warnings on the product packaging. In 2016, Chile was the first country with a mandatory FOP warning label system for foods high in calories, added sugar, sodium, and saturated fat, with several stages of implementation through 2019 [10]. Peru and Uruguay are both in the implementation stage of very similar warning label regulations [11,12], and Brazil’s health regulatory agency recently issued a public consultation for a similar FOP system [13]. While most of these regulations have yet to be evaluated, lab-based studies indicate that FOP warning labels attract consumers’ attention, help them identify unhealthy products, and reduce intentions to purchase and consume food [14,15,16,17,18]. A recent study in Chile found that mothers understand the warning labels and use the labels to make healthier food purchasing decisions for their children [19]. In addition to effects on consumer behavior, FOP warning labels may also encourage manufacturers to improve the nutritional qualities of their food in order to meet the nutrition criteria and thereby avoid the negative FOP labels [20,21].

Currently in Colombia, it is unknown what proportion of packaged foods and beverages available in the marketplace would meet the criteria for FOP warning labels. Considering that a bill to implement warning labels is currently under consideration in the Colombian Congress, it is essential to understand the scope of packaged products that would be subjected to regulation. Several nutrient profile models are currently under review by the Ministry of Health and Social Protection of Colombia, with discussions underway of applying these nutrient profile models to implement a new food labeling system. A nutrient profile model is defined as “the science of categorizing foods according to their nutritional composition for reasons related to preventing disease and promoting health” [22], which can then be used to identify foods subject to various policies such as an FOP warning label system [23]. Thus, it is critical to understand what proportion of the Colombian food supply would be subject to warning labels under different nutrient profile models.

The aim of the present study was to estimate what percentage of packaged food and beverage products currently available for purchase in Colombia would be subject to FOP warning labels under two different nutrient profile models: (a) the Pan American Health Organization model (PAHO model), and (b) the nutrient profile model used in the third phase of the Chilean Law on Food Labeling and Advertising (Chilean model).

## 2. Materials and Methods

This study was exempt from institutional review board approval at Javeriana University, Bogotá, Colombia, and the University of North Carolina at Chapel Hill, Chapel Hill, NC, USA.

### 2.1. Data Collection

This study was cross-sectional. Nutritional information for packaged foods and beverages was collected between August and November 2016, in 16 supermarkets from the five largest retail chains of Bogotá at the time of the study. Data were collected on all products available for purchase during this time regardless of the product’s country of origin. Supermarkets were located in neighborhoods of high, medium and low socioeconomic status, according to the criteria defined by the Major Office of Bogotá [24]. An agreement between the School of Medicine of Javeriana University and each retail chain was signed prior to data collection and permits were obtained in the selected supermarkets. Data were collected by 10 fieldworkers who received standardized training on data collection and entry [25]. Photos were taken on all sides of each product to capture the following: barcode, nutritional facts (i.e., energy, sodium/salt, total sugar, total fat, saturated fat, and trans fat), ingredients, product name, brand, manufacturer, and package size.

Study data were collected and managed using REDCap (Research Electronic Data Capture) hosted at the University of North Carolina at Chapel Hill [26]. Trained nutritionists viewed photos and entered information into a REDCap survey specifically designed for this study.

Based on their nutritional content and typical consumption in the Colombian diet, all products were classified into one of the following 12 food categories: beverages (e.g., softs drinks and teas with caloric or non-nutritive sweeteners); bread and bakery products (e.g., pastry, packaged bread); cereal products (e.g., breakfast cereals, whole grain cereals); convenience foods (e.g., ready to eat foods such as pizza); dairy (e.g., milk, yogurt, cheese); fish and seafood (e.g., tuna, shrimp); fruits, vegetables, nuts and legumes (e.g., bananas, canned beans); meats (e.g., unprocessed beef, sausages); sauces, dressings and spreads (e.g., mayonnaise, Chinese sauce); snack foods (e.g., packaged chips, microwave popcorn); sweets (e.g., candies, gummies); and miscellaneous (e.g., almond-based beverages, legume-based beverages, other foods for specific dietary uses). For beverage products sold in gram format, registered dietitians assigned a reconstitution factor to calculate the nutrient densities per 100 mL. Food products were evaluated in an ‘as purchased’ format only. Infant formula and culinary ingredients were excluded from analyses, including cooking oils, butter, salt, honey, sugar, and sweeteners.

### 2.2. Nutrient Profiling Systems

Two nutrient profiling models were chosen for comparison: the Pan American Health Organization (PAHO) model, and the nutrient profile established under the third phase of the Chilean Law on Food Labeling and Advertising (Chilean model). These models were examined because they are the most common systems in use or being discussed for food policy in Latin American countries, including Colombia. Nutrient criteria algorithms were developed according to the food classification criteria of each model. Each food item was individually classified according to nutrient criteria. First, each product was identified as containing added sugar, added salt/sodium, added saturated fat or non-nutritive sweeteners based on searching for keywords included in the list of ingredients. Briefly, keywords for added sugar included sugar, honey, syrups, maltodextrin, glucose, and fructose. Keywords for added salt/sodium included salt and sodium chloride. Keywords for added saturated fat included oils, butter, and animal and vegetable fats. Keywords for non-nutritive sweeteners included aspartame, saccharin, sucralose, cyclamate, acesulfame k, stevia, polydextrose, maltitol, mannitol, isomaltose, and neotame. All keyword searches were conducted using Spanish ingredient terms. Then, the overall nutrient level was assessed using the nutrient profile models as described below.

### 2.3. PAHO Model

The PAHO model and NOVA classification were considered as references to define products as unprocessed or minimally processed foods, and processed and ultra-processed foods [27,28]. Products were considered processed/ultra-processed if ingredient lists reported the presence of any added sugar, salt/sodium, saturated fat, and/or non-nutritive sweetener. Products considered unprocessed or minimally processed were considered unregulated. Only processed and ultra-processed foods were eligible to be classified as regulated based on nutrient thresholds.

Among processed/ultra-processed foods, a product was considered regulated under the PAHO model if one or more of the following criteria were met: (1) the presence of an added sugar ingredient and ≥10% of total energy contributed by free sugars; (2) ≥30% of total energy contributed by total fat; (3) ≥10% of total energy contributed by saturated fat; (4) ≥1% of total energy contributed by trans fat; (5) ≥1 mg of sodium per 1 kcal; or (6) the presence of a non-nutritive sweetener ingredient. Note that added sugars and free sugars include all monosaccharides and disaccharides added to foods by the manufacturer; free sugar additionally includes all naturally occurring sugars in honey, syrups, and non-intact (e.g., juiced or pureed) fruit and vegetables [28]. Total sugars include all added and free sugars, in addition to sugars that naturally occur in dairy products and intact fruit and vegetables. Free sugars were not reported on product packaging; thus, a free sugar factor (range 0 to 1) was assigned by registered dietitians multiplied by the total sugar value reported on the product packaging to estimate the value of free sugars for each product, following the algorithm developed by the Expert Consultation Group of PAHO Nutrient Profile Model [28].

### 2.4. Chilean Model

The Chilean nutrient profile model includes three phases, each increasingly more stringent. The current study applied the third and final phase of the Chilean nutrient profile model [10]. A product was considered regulated under the Chilean model if one or more of the following criteria were met: (1) it contained an added sugar ingredient and nutrient density of total sugar >10 g per 100 g, or >5 g per 100 mL; (2) it contained an added saturated fat ingredient and nutrient density of saturated fat >4 g per 100 g, or >3 g per 100 mL; (3) it contained an added sodium ingredient and nutrient density of sodium >400 mg per 100 g, or >100 mg per 100 mL; or (4) it contained an added sugar, saturated fat, or sodium ingredient and nutrient density of total energy >275 kcal per 100 g, or >70 kcal per 100 mL.

### 2.5. Statistical Analysis

First, the mean nutritional content of the food and beverage categories was examined. Then, the percentage of foods and beverages that met the criteria for each nutrient threshold, as well as the overall regulation status, was compared under the PAHO model and the Chilean model. Data analyses were performed using SAS version 9.4 (SAS Institute, Cary, NC, USA).

## 3. Results

Data were collected for 8948 products. Products were excluded from analyses if more than one nutrition facts panel was present (*n* = 144), the package included a multipack of different products (*n* = 1198), if they were culinary ingredients (*n* = 511), or if nutrient information was missing (*n* = 387). The final sample included 6708 products.

Table 1 presents the mean and standard deviations of nutrient densities by food category for total sugar, free sugar, total fat, saturated fat, trans fat, sodium, and energy per 100 g or 100 mL of product. Not surprisingly, the highest values for total sugar and free sugar were for sweets (both 46.1 g) and lowest in meats (both 0.4 g). Nutrient densities for total sugar and free sugar were fairly similar, with the exception of fruits, vegetables, nuts and legumes (20.2 g and 12.4 g, respectively), as well as dairy products (11.4 g and 8.2 g, respectively). Saturated fat densities were highest among sweets (10.3 g), followed by snack foods (8.8 g). The mean nutrient density for trans fats was less than 1 g in all food categories. Convenience foods had the highest nutrient density for sodium (2246.2 mg) followed by sauces, dressings and spreads (1108.4 mg). The highest mean energy density was found among snack foods (485 kcal) and sweets (461 kcal), with the lowest mean energy density for beverages (29 kcal). 

The proportion of products meeting the criteria for each nutrient threshold is presented for the PAHO model (Table 2) and Chilean model (Table 3). Based on the PAHO model, 80.2% of food and beverage products meet the criteria for regulation compared to 66.4% under the Chilean model. Among all products under the PAHO model, 41.8% are above the threshold for free sugar, 39.9% for total fat, 37.0% for saturated fat, 1.8% for trans fats, and 38.1% are above the threshold for sodium. In addition, 16.0% of products contain non-nutritive sweeteners. Among all products under the Chilean model, 34.2% are above the threshold for total sugar, 23.2% for saturated fat, 29.7% for sodium, and 37.0% are above the threshold for energy. By the food categorization under the PAHO model, the proportion of products meeting the criteria for free sugar was highest among sweets (74.4%) followed by beverages (64.1%). Under PAHO, the majority of products meet the saturated fat criteria for meats and snack foods (78.5% and 70.2%, respectively), and the sodium criteria for meats (90.0%) and convenience foods (80.2%). Under the Chilean model, the majority of sweets are above the threshold for total sugar (72.3%), the majority of snack foods are above the threshold for saturated fat (72.4%), and high proportions of meats, snack foods, and sauces, dressings and spreads are above the threshold for sodium (82.5%, 61.8%, 60.1%, respectively). High energy densities are found among most products in snack foods, sweets, and bread and bakery products (89.3%, 81.1%, and 69.1%, respectively).

The comparison of the proportion of products meeting regulation for each nutrient profile model by each food category is presented in Figure 1. Regarding the PAHO model, meats had the highest regulated proportion (97.3%) followed by sweets (95.6%). The lowest proportions were found in cereal products and miscellaneous (47.3% and 69.9%, respectively). The highest proportions of foods and beverages to be regulated by the Chilean model were snack foods (92.0%) followed by meats (87.5%). Miscellaneous and fish and seafood had the lowest proportions (39.5% and 39.0%, respectively). The proportion of products meeting regulation criteria was similar under PAHO and Chilean models for bread and bakery products (83.9% vs. 80.6%), cereal products (47.3% vs. 48.2%), sauces, dressings and spreads (81.4% vs. 79.9%), snack foods (92.4% vs. 92.0%). and somewhat similar for meats (97.3% vs. 87.5%).

Products were much more likely to meet regulation criteria under the PAHO model compared with the Chilean model for beverages (80.2% vs. 42.9%), convenience foods (87.7% vs. 63.4%), dairy (88.9% vs. 71.3%), fish and seafood (90.0% vs. 39.0%), fruits, vegetables, nuts and legumes (75.8% vs. 55.8%), and miscellaneous (69.9% vs. 39.5%). More beverages meet the regulation criteria under the PAHO free sugar criteria (64.1%) compared with Chilean total sugar criteria (41.7%). Products are more likely to meet the sodium criteria for PAHO compared with the Chilean model for convenience foods (80.2% vs. 56.8%), fish and seafood (79.5% vs. 36.3%), fruits, vegetables, nuts and legumes (29.5% vs. 18.7%) and miscellaneous (31.1% vs. 7.1%), and more likely to be above the saturated fat threshold for dairy (68.5% vs. 30.1%).

Finally, 41.0% of beverages and 28.0% of foods meet the criteria for regulation based on the presence of non-nutritive sweeteners in the PAHO model.

## 4. Discussion

Currently, the Colombian Congress is considering a bill to implement FOP warning labels on packaged foods and beverages. However, to our knowledge, no detailed studies have been conducted to provide information on the proportion of products in the Colombian food supply that would be regulated under various FOP systems. In this study, two nutrition profile models were examined among 6708 foods available for purchase in 16 of the largest supermarket retailers across Bogotá. Overall, 80.2% of the packaged food and beverage products meet the regulation criteria under the PAHO model, and 66.4% under the Chilean model. Both nutrient profile systems indicate that the majority of packaged foods sold in Colombian supermarkets contain excess critical nutrients of which major world health organizations including PAHO and the WHO recommend limited consumption [6,29]. These results are similar to several studies conducted in Mexico and Honduras [30,31]. Consequentially, this suggests that if a FOP warning label system similar to that of Chile, Peru, or Uruguay was to be implemented in Colombia, then the majority of packaged products should receive at least one FOP warning label.

There are differences in the capacity of the selected models to identify the processed and ultra-processed foods that would be regulated. More products would be subjected to regulation under the PAHO model than the Chilean model, particularly among food categories not necessarily considered part of an unhealthy diet, such as dairy, fish and seafood, and fruits, vegetables, nuts and legumes. There were also striking differences for some categories for other nutrients as well. For example, nearly 69% of dairy products and 79% of meat products exceed saturated fat thresholds under PAHO, but only 30% of dairy products and 26% of meats exceed these thresholds for the Chilean model. For fruits and vegetables, 76% of total products under PAHO were regulated while only 56% were under the Chilean model. Interestingly, the percentage regulated for sugar was similar in both systems (38% for PAHO and 34% for Chile), but there was a relatively high percentage regulated under PAHO for total fat (27%) and sodium (30%), whereas for Chile, 20% were regulated for energy and 19% for sodium.

Some of the differences between the Chilean and PAHO models may reflect differences in the use of a nutrient density vs. volume-based approach for classifying foods. Specifically, the PAHO model is mainly based on the energy density of selected nutrients (i.e., nutrients per calorie), while the Chilean model calculates the content of nutrients per 100 g or milliliters of total product (i.e., nutrients per volume). Subsequently, under PAHO, products low in calories could still be subjected to regulation because they could have a relatively high nutrient-to-calorie ratio. For example, under the PAHO model, sodium criteria are based on a sodium/calorie ratio (i.e., ≥1 mg of sodium per 1 kcal), whereas under the Chilean model, the criteria are based on sodium density (i.e., an added sodium ingredient and a nutrient density of sodium >400 mg per 100 g, or >100 mg per 100 mL). Therefore, under PAHO, products such as frozen vegetables which contain few calories but do contain sodium could potentially exceed sodium thresholds, despite containing low levels of sodium overall.

Because of this nutrient-per-calorie approach, under the PAHO model, products with relatively high energy densities might be less likely to exceed nutrient densities for nutrients such as sugar or sodium, because it will be difficult to exceed the percentage of calorie thresholds. For example, a peanut butter spread with 15 g of sugar per 100 g of product might only have 9% of calories from sugar, making it ineligible for the sugar warning (although it would receive a fat warning). Other potential categories where this could be an issue could include nut spreads, snack foods, sweets and convenience foods. Thus, one potential concern about the PAHO model is that the likelihood of a product being regulated depends on the nutrient density of the model. On the other hand, the current study showed that, in general, more products were classified as regulated under the PAHO model compared to the Chilean model, and so this issue may not apply to many products. More specific analyses looking at foods and beverages within each category could be useful to understand the healthfulness of products classified as regulated or unregulated by each nutrient profile model.

Another notable difference between the two models was found among beverages. Nearly twice as many beverages met regulation criteria under the PAHO model (78%) compared to the Chilean model (42%). This is due to a greater proportion of beverages exceeding sugar thresholds for PAHO (64%) compared to the Chilean model (43%), potentially due to differences in how sugar is defined (the PAHO model uses free sugar, while the Chilean model uses total sugar). Additionally, beverages containing non-nutritive sweeteners are eligible to be regulated under PAHO (43% of beverages), but not the Chilean model. For both models, one limitation of the current approach is that all beverages are grouped together, and thus it is not possible to understand differences in sugar levels across beverage subcategories. It is likely that sugar-sweetened beverages such as carbonated soft drinks and fruit drinks have a much higher proportion of products exceeding sugar criteria than other beverages, such as diet soft drinks or waters, but the current study was unable to observe differences within this category due to the aggregated grouping approach. More research will be needed to understand the nutritional profile of beverage subcategories

From a nutritional perspective, the PAHO model includes a regulation based on excess total fat, which is not aligned with current evidence suggesting that it is the type of fat, not total fat, that matters for health [32]. Similarly, the evidence on the negative health impact as well as the benefits of non-nutritive sweeteners as a replacement for caloric sweeteners is currently unclear [33]. On the other hand, the Chilean model does not include trans fat, which is important to reduce in the food supply in order to prevent cardiovascular diseases [34]. It is important to note that Chile already had previous regulations to reduce trans fats which were implemented before the labeling and advertising laws [35]. The mean nutrient density for trans fats was less than 1 g in all food categories in this study; however, under the PAHO model, 1.8% of the processed foods exceed the threshold for trans fats. Although this percentage is lower than for others Latin American countries [36], special attention by the Colombian public health institutions is required to reduce it.

These results should be evaluated with consideration several limitations of this study. First, because of the significant socioeconomic and cultural differences among the Colombian regions, which may influence the food supply in each region, the findings of this study are restricted to Bogotá. However, one strength of the study is that data were collected among supermarkets located across a range of socioeconomic-status neighborhoods, and these supermarkets represent the top ten Colombian food retailers in sales in the same period of the study {Euromonitor 2016}. Second, this study does not include foods bought in bulk, un-packaged foods (e.g., fruits, vegetables, meats, cheese, bread), or foods available for purchase outside of supermarkets in its analysis, since the information for this study was obtained from packaged foods and beverages with barcodes and nutrient information offered in the supermarkets. However, because the FOP warning label system will likely apply mainly to packaged foods and beverages, this study sample does represent most of the products that might be affected by such a regulation.

Third, the study uses broad groupings of food categories, and thus the heterogeneity in what types of products would be regulated within each category cannot be clearly understood. For example, within the beverage category, fruit drink sales are increasing. These beverages tend to contain a great deal of added sugar and thus are more likely to be regulated; however, differences between fruit drinks and other types of beverages are not under the scope of the current study. This situation could be similar in other categories. For example, within the cereal products category, breakfast cereals may contain a much greater level of high-sugar products that are likely to be regulated, whereas plain cereals such as rice or pasta would be less likely to be regulated. Thus, more research is needed to understand what the overall sugar levels and percentage of products would be regulated within heterogeneous categories, such as beverages and cereals.

Finally, Colombia does not currently require that information on the free sugar content of products be included in the nutrition facts panel. Free sugar estimates were assigned and reviewed by registered dietitians, but the actual free sugar values are unknown. As a large proportion of products in this sample included excess free sugar, it is important that the Colombian nutrition facts panel includes mandatory reporting of free sugar to provide this information for the consumers, particularly if future regulations include criteria corresponding to levels of free sugar.

## 5. Conclusions

This study found that the majority of the packaged food supply in Colombia exceeded critical nutrient thresholds using the PAHO and Chilean nutrient profile models. These results suggest that a warning label regulation that applies these nutrient thresholds could have a significant impact on the Colombian food supply, as well as the foods and beverages which the Colombian population purchases and consumes. Additional research on these nutrient profile models, including evaluations of existing labeling policies that apply these nutrient profile models, will be needed to inform population-level strategies to maintain healthy diets and reduce obesity in the Colombian population.

## Figures and Tables

**Figure 1 nutrients-11-01011-f001:**
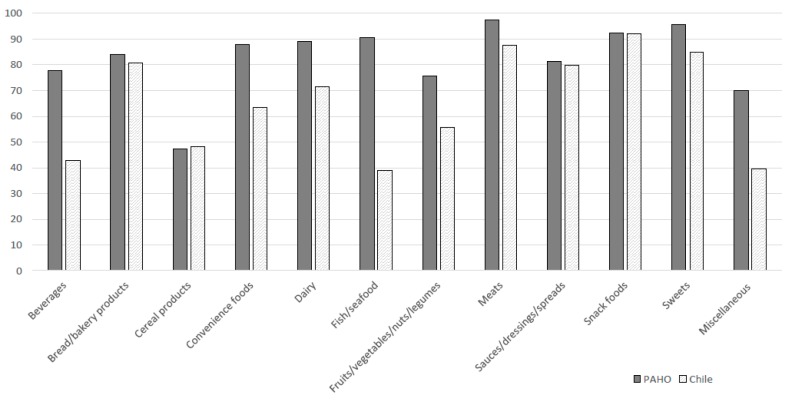
Proportion of packaged food and beverage products meeting the criteria for regulation under PAHO and Chilean nutrient profile models.

**Table 1 nutrients-11-01011-t001:** Mean (standard deviation) of nutrient densities (per 100 g or per 100 mL) by food category for packaged food and beverage products available for purchase in Bogotá, Colombia (*n* = 6708).

Category	Products *n* (%)	Total Sugar (g)	Free Sugar (g)	Total Fat (g)	Saturated Fat (g)	Trans Fat (g)	Sodium (mg)	Energy (kcal)
		Mean (SD)	Mean (SD)	Mean (SD)	Mean (SD)	Mean (SD)	Mean (SD)	Mean (SD)
Beverages	604 (9%)	6.1 (4.8)	5.2 (4.3)	0.1 (0.7)	0.0 (0.2)	0.0 (0)	12 (32.8)	29 (24)
Bread and bakery products	923 (14%)	16.2 (16.4)	16.2 (16.4)	11.0 (14.4)	5.0 (5.9)	0.1 (0.5)	421.4 (738)	366 (184)
Cereal products	819 (12%)	11.5 (13.4)	11.5 (13.4)	3.4 (5.4)	1.0 (2.2)	0.0 (0.1)	178 (281)	332 (143)
Convenience foods	424 (6%)	3.7 (5.0)	3.7 (5.0)	7.4 (14.9)	2.9 (4.0)	0.1 (0.7)	2246.2 (4314)	228 (214)
Dairy	1026 (15%)	11.5 (14.7)	8.2 (11.4)	12 (18.3)	6.6 (7.6)	0.2 (1.6)	208.6 (411)	193 (169)
Fish and seafood	190 (3%)	0.6 (1.6)	0.6 (1.6)	7.1 (5.5)	2.4 (13.3)	0.0 (0)	419.9 (600)	153 (58)
Fruits, vegetables, nuts and legumes	739 (11%)	20.2 (24.9)	12.4 (16.7)	10.2 (18.5)	2.0 (4.5)	0.1 (0.5)	368.4 (775)	220 (215)
Meats	451 (7%)	0.4 (1.4)	0.4 (1.4)	15.1 (16.4)	5.5 (5.1)	0.0 (0.2)	961.9 (694)	215 (110)
Sauces, dressings and spreads	636 (9%)	13.7 (23.4)	13.7 (23.4)	16.3 (25.0)	5.6 (10.6)	0.1 (1)	1108 (1491)	236 (243)
Snack foods	225 (3%)	4.8 (10.4)	4.1 (10.3)	23.2 (10.3)	8.8 (5.6)	0.2 (1.4)	643.3 (443)	485 (136)
Sweets	433 (6%)	46.1 (31.4)	46.1 (31.4)	19.6 (16.9)	10.3 (10.0)	0.1 (0.6)	73.1 (253)	461 (270)
Miscellaneous	238 (4%)	5.9 (9.5)	5.9 (9.5)	9.8 (16.2)	2.6 (6)	0.7 (3.8)	106.9 (212)	191 (204)

**Table 2 nutrients-11-01011-t002:** Number and proportion of packaged food and beverage products meeting criteria for regulation under the Pan American Health Organization nutrient profile model (PAHO model), by food category.

Category	Regulated ^1^		Meets Criteria for Nutrient Threshold											
			Free Sugar ^2^		Total Fat ^3^		Saturated Fat ^4^		Trans Fat ^5^		Sodium ^6^		NNS ^7^	
	*n*	%	*n*	%	*n*	%	*N*	%	*n*	%	*n*	%	*n*	%
Beverages	470	77.8	387	64.1	6	1.0	7	1.2	0	0	53	8.8	249	41.2
Bread and bakery products	774	83.9	447	48.4	380	41.2	375	40.6	16	1.7	412	44.6	148	16.0
Cereal products	427	47.3	303	37.0	48	5.9	68	8.3	4	0.5	144	17.6	162	18.0
Convenience foods	372	87.7	61	14.4	168	39.6	161	38.0	18	4.3	340	80.2	22	5.2
Dairy	1045	88.9	588	57.3	616	60.0	703	68.5	45	4.4	261	25.4	218	18.6
Fish and seafood	172	90.5	3	1.6	120	63.2	60	31.6	0	0	151	79.5	6	3.2
Fruits, vegetables, nuts and legumes	560	75.8	277	37.5	200	27.1	114	15.4	8	1.1	218	29.5	62	8.4
Meats	439	97.3	7	1.6	360	79.8	354	78.5	7	1.6	406	90.0	34	7.5
Sauces, dressings and spreads	518	81.4	300	47.2	253	39.8	207	32.6	5	0.8	362	56.9	32	5.0
Snack foods	208	92.4	16	7.1	181	80.4	158	70.2	4	1.8	134	59.6	8	3.6
Sweets	414	95.6	322	74.4	249	57.5	230	53.1	4	0.9	3	0.7	123	28.4
Miscellaneous	186	69.9	93	39.1	93	39.1	47	19.8	8	3.4	74	31.1	50	18.8
All food and beverages	5585	80.2	2804	41.8	2674	39.9	2484	37.0	119	1.8	2,558	38.1	1114	16.0

^1^ Regulated if the product is classified as processed food and met the criteria for at least one nutrient threshold. ^2^ Product included an added sugar ingredient and ≥10% total energy is contributed by free sugar. ^3^ ≥30% of total energy is contributed by total fat. ^4^ ≥10% of total energy is contributed by saturated fat. ^5^ ≥1% of total energy is contributed by trans fat. ^6^ ≥1 mg of sodium per 1 kcal. ^7^ Product includes a non-nutritive sweetener (NNS) in the ingredient list.

**Table 3 nutrients-11-01011-t003:** Number and proportion of packaged food and beverage products meeting the criteria for regulation under the Chilean model, by food category.

Category	Regulated ^1^		Meets Criteria for Nutrient Threshold							
			Total Sugar ^2^		Saturated Fat ^3^		Sodium ^4^		Energy ^5^	
	*n*	%	*n*	%	*n*	%	*n*	%	*n*	%
Beverages	259	42.9	252	41.7	1	0.2	5	0.8	3	0.5
Bread and bakery products	744	80.6	400	43.3	334	36.2	331	35.9	638	69.1
Cereal products	395	48.2	286	34.9	58	7.1	115	14.0	372	45.4
Convenience foods	269	63.4	15	3.5	103	24.3	241	56.8	137	32.3
Dairy	731	71.3	481	46.9	309	30.1	179	17.5	272	26.5
Fish and seafood	74	39.0	-	-	7	3.7	69	36.3	1	0.5
Fruits, vegetables, nuts and legumes	412	55.8	252	34.1	72	9.7	138	18.7	144	19.5
Meats	395	87.5	-	-	118	26.2	372	82.5	73	16.2
Sauces, dressings and spreads	508	79.9	209	32.9	159	25.0	382	60.1	232	36.5
Snack foods	207	92.0	22	9.8	163	72.4	139	61.8	201	89.3
Sweets	368	85.0	313	72.3	200	46.2	1	0.2	351	81.1
Miscellaneous	94	39.5	65	27.3	29	12.2	17	7.1	56	23.5
All food and beverages	4456	66.4	2295	34.2	1553	23.2	1989	29.6	2480	37.0

^1^ Regulated if the product met the criteria for at least one nutrient threshold. ^2^ Product included an added sugar ingredient and had a total sugar >10 g per 100 g or >5 g per 100 mL. ^3^ Product included an added saturated fat ingredient and saturated fat >4 g per 100 g or >3 g per 100 mL. ^4^ Product included an added sodium ingredient and sodium >400 mg per 100 g or >100 mg per 100 mL ^5^ Product included added sugar, saturated fat, or sodium ingredient and had a total energy >275 kcal per 100 g or >70 kcal per 100 mL. Dashes indicate that no products met the nutrient threshold.

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
