# Peer review of "Nutrition Quality of Packaged Foods in Bogotá, Colombia: A Comparison of Two Nutrient Profile Models"

_nutrients, 2019, doi:10.3390/nu11051011_

Round 1
Reviewer 1 Report
The overall organization and presentation of the study, the results and the conclusions drawn are adequate.
Small grammatical errors
I believe that in some cases "would" is better to be replaced by "should" (line 236 and may be line 22). Also in line 23 and line 69 replace "be subject" to "be subjected to"
Line 290 replace "data was" with "data were"
Line 302 replace "do not present" with "does not present"
Author Response
Bogotá DC, April 25, 2019
Doctor
Prof. Dr. Lluis Serra-Majem
Editor-in-Chief
Nutrients
On behalf of the authors I want to thank you and the reviewers for the helpful comments and thorough review. We have reviewed each comment point by point and addressed these comments below.
Best regards
Mercedes Mora-Plazas
Departamento de Nutrición Humana.
Universidad Nacional de Colombia, Bogotá, Colombia.
Reviewer #1
| Comments | Our response |
1. | “The overall organization and presentation of the study, the results and the conclusions drawn are adequate. Small grammatical errors” | We have proof-read the paper an additional time.
|
2. | “I believe that in some cases "would" is better to be replaced by "should" (line 236 and may be line 22).” | We have made this correction.
|
3. | “Also in line 23 and line 69 replace "be subject" to "be subjected to"” | We have made this correction. |
4. | “Line 290 replace "data was" with "data were"” | We have made this correction. |
5. | “Line 302 replace "do not present" with "does not present"” | We have changed this sentence, so it no longer includes this phrase. |

Reviewer 2 Report
Very good information for decision making. I suggest optimizing the edition of the results to make the displayed data understandable in the best way. At the end of the article, it is not clear to me which model is "similar" the proposal that already has the Colombian Congress, and which, of the two models studied, would propose to be taken into account for the benefit of the population. I think it's important to mention some implications for the industry.

Author Response
Bogotá DC, April 25, 2019
Doctor
Prof. Dr. Lluis Serra-Majem
Editor-in-Chief
Nutrients
On behalf of the authors I want to thank you and the reviewers for the helpful comments and thorough review. We have reviewed each comment point by point and addressed these comments below.
Best regards
Mercedes Mora-Plazas
Departamento de Nutrición Humana.
Universidad Nacional de Colombia, Bogotá, Colombia.
Reviewer #2 | ||
1. | “Very good information for decision making. I suggest optimizing the edition of the results to make the displayed data understandable in the best way. At the end of the article, it is not clear to me which model is "similar" the proposal that already has the Colombian Congress, and which, of the two models studied, would propose to be taken into account for the benefit of the population. I think it's important to mention some implications for the industry.” | Our response: We have removed the referral to the Colombian Congress at the end of the article, because the political situation of this bill is changing rapidly and may be out of date when the article is published. We don’t necessarily feel that our current study results provide a compelling rationale to recommend either nutrient profile model. Future research will be needed to understand at a finer level of detail the healthfulness of products within categories under the two nutrient profile models, as we note. In addition, evaluations of existing laws under each model will be needed to understand which is better for improving population health. We have now noted this in the last sentence of the conclusion. Finally, in the introduction, we mention the most relevant point for the industry, which is that these laws may incentivize reformulation. |
2. | “It is necessary write the percentage in Chilean Model (lines 28-29)” | The abstract includes the percentage of foods that met criteria for regulation according to Chilean model. These are the lines: “Results: 80.2% (PAHO model) to 66.4% (Chilean model) of foods met criteria for regulation” (lines 28-29). |
3. | “Improve keywords with MeSH terms.” | We have amended the keywords to better reflect the study content. |
4. | “Type, study design?”
| We have added a sentence under “data collection” which notes that this study was cross-sectional in design. |
5. | Sample: food products with Colombian origin?
| The study included all products, regardless of country of origin. We now include the following statement: Data were collected on all products available for purchase during this time regardless of product country of origin. |
6. | I would suggest adding a line of total to the table one, and adding the percentage of each category.
| We have added the percent of products in Table 1. |
7. | Very large paragraph, I would suggest dividing to improve reading (lines 235-255). | We have divided this paragraph. |
8. | Very large paragraph, I would suggest dividing to improve reading (lines 288-306)
| We have divided this paragraph. |
9 | Review how references should be cited, depending on the type of work, e.g.: Correct (line 391) Incomplete (line 405)
| We have adjusted the reference format following the instruction of the journal.
|
10 | Standardize Kcal or kcal throughout the document
| We have standardized kcal throughout the document. |
11 | Separate the units of the abbreviations, for example 100 ml and not 100ml, throughout the document
| We separated the units of the abbreviations throughout the document. |
12 | Uniformize the use of decimals throughout the document. | We have uniformize the use of decimals throughout the document. |
13 | Uniformize the use of N - n throughout the document.
| We uniformize the use of N throughout the document. |
14 | Authors could consider article references of Nutrients 2018 (DOI: 10.3390/nu10060737) and the Spanish Journal of Nutrition and Dietetics 2018 (DOI: http://dx.doi.org/10.14306/renhyd.22.2.413), both of them from Latin American region. | We have included a brief mention about the results of these studies. (Lines: 254-255). |
